# Normal manual straight ahead pointing in Complex Regional Pain Syndrome

**Axel D. Vittersø**[1,2,3,4]*, **Gavin Buckingham**[3], **Antonia F. Ten Brink**[1,2],
**Monika Halicka**[1,2], **Michael J. Proulx**[2,5], **Janet H. Bultitude**[1,2]

1 Centre for Pain Research, University of Bath, Bath, Somerset, United Kingdom, 2 Department of Psychology, University of Bath, Bath, Somerset, United Kingdom, 3 Department of Sport & Health Sciences, University of Exeter, Exeter, Devon, United Kingdom, 4 Department of Psychology, Oslo New University College, Oslo, Norway, 5 Centre for Real and Virtual Environments Augmentation Labs, Department of Computer Science, University of Bath, Bath, Somerset, United Kingdom

* axel.vitterso@oslonh.no

**Data Availability Statement:** All relevant data are within the manuscript and/or available from https:// osf.io/t9j52/.

**Funding:** ADV received funding from the GW4 BioMed Medical Research Council Doctoral

## Abstract

There is evidence to suggest that people with Complex Regional Pain Syndrome (CRPS) can have altered body representations and spatial cognition. One way of studying these cognitive functions is through manual straight ahead (MSA) pointing, in which participants are required to point straight ahead of their perceived body midline without visual feedback of the hand. We therefore compared endpoint errors from MSA pointing between people with CRPS (n = 17) and matched controls (n = 18), and examined the effect of the arm used (Side of Body; affected/non-dominant, non-affected/dominant). For all participants, pointing errors were biased towards the hand being used. We found moderate evidence of no difference between Groups on endpoint errors, and moderate evidence of no interaction with Side of Body. The differences in variability between Groups were non-significant/inconclusive. Correlational analyses showed no evidence of a relationship between MSA endpoint errors and clinical parameters (e.g. CRPS severity, duration, pain) or questionnaire measures (e.g. body representation, "neglect-like symptoms", upper limb disability). This study is consistent with earlier findings of no difference between people with CRPS and controls on MSA endpoint errors, and is the first to provide statistical evidence of similar performance of these two groups. Our results do not support a relationship between clinical or self-reported measures (e.g. "neglect-like symptoms") and any directional biases in MSA. Our findings may have implications for understanding neurocognitive changes in CRPS.

## Introduction

Complex Regional Pain Syndrome (CRPS) is pathological pain condition characterised by motor deficits, and autonomic symptoms [1, 2]. This condition can also be accompanied by neuropsychological changes [for reviews, see 3, 4], such as changes in spatial perception, which might be considered neglect-like [3, 5], although alternative interpretations exist [6–8]. Recently we showed that people with CRPS do not show a consistent visuospatial attention bias [8, 9]. However, some studies that have suggested biases in the representations of space (i.e. the mental knowledge or model of external space). For example, when asked to indicate

Training Partnership (1793344). AFTB was
supported by a Rubicon grant (019.173SG.019)
from the Netherlands Organisation for Scientific
Research (NWO). The funders had no role in study
design, data collection and analysis, decision to
publish, or preparation of the manuscript.

**Competing interests:** The authors have declared
that no competing interests exist.

when a visual target passes the point in front of the body midline in an otherwise darkened
room ("visual straight ahead" judgements; VSA), people with CRPS have shown a bias away
from the affected arm [10–14], a leftward bias [15] (i.e. pseudoneglect), or no bias [16, 17].
When performed in the dark, VSA is thought to reflect any lateral shifts in the representations
of the body midline and of external space relative to one's own position (i.e. egocentric spatial
representations) [14, 18]. However, when performed in light conditions, the availability of
other visual cues makes it possible to recruit allocentric representations (i.e. the representation
of objects in space relative to each other). The observation of normal VSA judgments made by
people with CRPS in light conditions [14] therefore suggest that only egocentric, and not allo-
centric, representations of space are biased.

Manual straight ahead (MSA) is related to VSA, and involves pointing straight in front of
one's perceived body midline without visual feedback of the pointing arm's location (e.g. with
the eyes closed). MSA has previously been used to quantify directional biases of the perceived
body midline in an egocentric reference frame in healthy controls [e.g. 19–23], and people
with post-stroke neglect. Like VSA in dark conditions, MSA is thought to reflect egocentric
representations of space [24]. However, unlike VSA, MSA can also provide insights into pro-
prioceptive accuracy (i.e. in the pointing arm), and the ability to align the felt position of the
arm with the perceived direction of straight ahead. Two studies found that people with CRPS
were more variable than control participants when matching the position of their affected or
unaffected arm to externally-defined targets [25, 26], indicating impaired proprioception. Fur-
thermore, a recent study of people with CRPS found that their pointing accuracy depended on
the type of sensory information available (visual, proprioceptive, or visual-proprioceptive)
[27]. It could therefore be informative to use MSA to gain further insights into both spatial
representations and limb proprioception, and to evaluate whether there are also deficits in the
way that arm proprioceptive information is combined with egocentric spatial representations.

Even though six of the eight studies to investigate VSA in CRPS have reported evidence for
a directional bias, almost none of the studies that have evaluated MSA in CRPS have found
any such evidence. Two case studies of a woman with unilateral CRPS found that her MSA
was biased toward the affected side when using either hand [10, 11], suggesting hyperattention
towards (rather than "neglect" of) the affected limb. By contrast, MSA did not deviate from
zero for either hand when averaged across seven people with CRPS [17]. Larger group studies
also reported no MSA bias for either hand in people with CRPS compared to pain-free controls
[n = 17; 27], or other pain patients and pain-free controls [n = 20; 28]. However, since the
pointing errors were measured against external objects that was seen by the participant prior
to the task (respectively, a large protractor, and a vertical line marked on the well), these stud-
ies might have inadvertently encouraged the use of allocentric reference frames (i.e. if the par-
ticipant pointed to the remembered location of the 0˚ mark on the protractor, or the line on
the wall, rather than to straight ahead of their body midline). The apparent lack of MSA bias in
CRPS despite stronger evidence for VSA bias could indicate that any bias in the representation
of egocentric space is overcome when arm proprioception is involved in the task. However,
much of the current understanding of MSA in CRPS is limited by potential task confounds, or
small sample sizes. Furthermore, most of these studies did not analyse whether there were any
differences in the variability of pointing errors made by people with CRPS, which would pro-
vide insights into possible deficits in arm proprioception or in aligning felt arm position with
the spatial representation of straight ahead. The only exception is the study by Kolb and her
colleagues [28] which found no evidence of higher MSA variability in people with CRPS.
Finally, the existing studies only allow the conclusion that there is no evidence for a difference
in the MSA directional errors and variability of CRPS patients and controls: they do not pro-
vide evidence that they are equivalent.

To address these gaps, we used motion capture to sensitively measure MSA in people with upper limb CRPS-I and pain-free controls. We compared differences between Groups (CRPS, controls) and Side of Body (affected/non-dominant, non-affected/dominant) on MSA and its variability. If people with CRPS have a directional bias in their egocentric spatial representations, we would expect to see greater endpoint errors on MSA than for controls. We would also expect people with CRPS to show greater variability on MSA than controls if their proprioception is less precise. If such variability is present to similar extents for pointing with the affected and unaffected limb, this could indicate higher level problems with aligning felt limb position with egocentric spatial representations (rather than, for example, proprioceptive deficits due to peripheral changes). Previous research has found an association between VSA and "neglect-like symptoms" [15], although others do not find evidence for such a relationship [16]. We therefore explored the relationship between MSA, clinical data, and questionnaire measures to see if spatial biases were related to CRPS symptoms, "neglect-like symptoms", and/or disability. Finally, we complemented our frequentist analyses with Bayesian statistics to allow insights into the weight of evidence for the null hypothesis as well as the alternate hypothesis [29].

## Materials and methods

### Participants

We recruited 17 people with unilateral upper limb CRPS-I ($M_{age}$ = 53.53, $SD$ = 11.67; 16 female; 14 right-handed; Table 1); and 18 pain-free controls matched for age, sex, and handedness ($M_{age}$ = 54.17, $SD$ = 12.22; 17 female; 15 right-handed). We decided on our sample size pragmatically. The target sample size was based on the maximum number of people with CRPS we could feasibly recruit and test given financial and time constraints. The final sample size meant that the study was able to reliably detect a large effect of $\eta_p^2 \geq .21$, with an alpha of .05, and 80% power. Twelve participants met the Budapest research criteria for CRPS [1, 2], three met the clinical criteria, and two met the criteria for CRPS not otherwise specified. After completing the MSA task, all participants also took part in a study that aimed to characterise the process of sensorimotor adaptation [30]. The current study is reported separately because it was focused on assessing the presence of any systematic bias in MSA for people with CRPS, rather than the difference in the transfer of sensorimotor prism adaptation after-effects to spatial representations [e.g. 31]. Exclusion criteria for both groups were a history of brain injury, brain disorders, or psychiatric disorders. For safety reasons, we excluded people with a pacemaker, spinal cord stimulator or similar devices; or who were pregnant or breastfeeding. The study complied with the 2013 declaration of Helsinki and had ethical permission from the UK Health Research Authority (REC reference 12/SC/0557). Informed written consent was obtained.

### Procedure

All individuals provided informed written consent prior to participation. People with CRPS then went on to have an assessment of their sensory, vasomotor, sudomotor/oedema, and motor/trophic signs and symptoms of CRPS. The number of reported symptoms and observed signs were used to calculate a CRPS severity score [Table 1; 37]. To assess handedness, all participants completed the Edinburgh handedness inventory [38]. A score < -40 indicates left-handedness, a score > 40 indicates right-handedness, and any other score indicates ambidextrousness. Three people with CRPS were classed as left-handed, four as ambidextrous, and eight as right-handed. Two control participants were classed as left-handed, three as ambidextrous, and 11 as right-handed.

**Table 1. Clinical information for people with upper limb CRPS.**

| ID | CRPS Severity; Budapest criteria | Duration (months) | Current pain | Pain DETECT | CRPS BPD | DASH | TSK | NBQ | Inciting event | Medication | Comorbidities |
|---|---|---|---|---|---|---|---|---|---|---|---|
| UL1 | 13; R | 67 | 8 | 24 | 20 | 65.9 | 29 | 3.2 | Soft tissue injury of the hand | Co-codamol, etodolac, omeprazole, amitriptyline, sertraline | TMJ, FMS, IBS, migraine |
| UL2 | 5; C | 64 | 4 | 15 | 14 | 29.5 | 29 | 1.8 | Hand surgery | Aspirin, bisoprolol fumarate, levothyroxine sodium, ramipril, folic acid, methotrexate, statin, paracetamol | Frozen joints, arthrosis |
| UL3 | 10; R | 32 | 8 | 29 | 43 | 79.5 | 39 | 4.2 | None identified | Buprenorphine, gabapentin, naproxen, omeprazole, antihistamine, promethazine | FMS, migraine, PCOS, asthma |
| UL4 | 7; NOS | 99 | 2 | 21 | 7 | 31.8 | 27 | 1.2 | Elbow spiral fracture | Aspirin, felodipine, ramipril, paracetamol, lansoprazole | FMS |
| UL5 | 11; R | 93 | 2 | 11 | 16 | 43.2 | 20 | 1.6 | Soft tissue injury of the hand | Paracetamol, ibuprofen | |
| UL6 | 12; R | 74 | 9 | 30 | 36 | 77.3 | 41 | 3.2 | Shoulder surgery | Gabapentin, topiramate, zolmitriptan, paracetamol, ibuprofen, senna glycoside | Migraine, frozen shoulder |
| UL7 | 10; C | 79 | 2 | 22 | 15 | 31.8 | 21 | 2.0 | None identified | None | |
| UL8 | 6; NOS | 91 | 1 | 8 | | 11.4 | 29 | 2 | Wrist fracture | Pregabalin, amitriptyline, calcium carbonate | |
| UL9 | 11; R | 140 | 8 | 11 | 22 | 52.3 | 37 | 3.2 | Multiple hand fractures | Bisoprolol | |
| UL10 | 11; R | 39 | 10 | 19 | 29 | 63.6 | 41 | 3.6 | Elbow fracture | Amitriptyline, omeprazole | |
| UL11 | 11; R | 148 | 4 | 28 | 33 | 52.3 | 31 | - | Wrist fracture | Pregabalin, amitriptyline, co-codamol, paracetamol | Low mood |
| UL12 | 10; R | 16 | 8 | 12 | 22 | 38.6 | 40 | 3.0 | Wrist fracture | Amitriptyline | Cartilage damage in knee (Left) |
| UL13 | 11; R | 43 | 5 | 17 | 21 | 54.5 | 26 | 2.2 | Surgery for dislocated shoulder | Morphine sulphate, pregabalin, propranolol | Migraines, PCOS |
| UL14 | 9; C | 59 | 6 | 10 | 13 | 36.4 | 38 | 1.6 | Soft tissue injury of the wrist | Co-codamol, amitriptyline, pregabalin | |
| UL15 | 14; R | 39 | 5 | 24 | 32 | 77.3 | 40 | 3.4 | | Nortriptyline, paracetamol, aminophylline, budesonide, formoterol fumarate dihydrate, salbutamol sulphate | Asthma |
| UL16 | 12; R | 14 | 6 | 26 | 33 | 59.1 | 52 | 5.6 | Multiple wrist fractures | Pregabalin, paracetamol | Diabetes |
| UL17 | 8; R | 138 | 6 | 16 | 7 | - | - | 1.0 | Forearm fracture | Amitriptyline, tramadol, amlodipine | FMS |
| *M* (*SD*) | 10.06 (2.41) | 72.65 (41.62) | 5.53 (2.74) | 19.00 (7.17) | 22.69 (10.66) | 50.28 (19.74) | 33.75 (8.58) | 2.68 (1.22) | | | |

BDP = Body perception disturbance score [32]. C = Clinical criteria for CRPS met. DASH = The Disabilities of the Arm, Shoulder and Hand questionnaire [33]. FMS = fibromyalgia syndrome. IBS = Irritable bowel syndrome. NBQ = Neurobehavioral questionnaire ("neglect-like symptoms") [34, 35]. NOS = CRPS not otherwise specified. PCOS = Polycystic ovary syndrome. TMJ = Temporomandibular joint syndrome. TSK = Tampa scale of kinesiophobia [36]. R = Research criteria for CRPS met.— = not measured.

People with CRPS then completed questionnaire measures (neuropathic-type pain, body representation disturbance, "neglect-like symptoms", upper limb disability, fear of movement [32, 34–36, 39]). Neuropathic type pain was assessed by the painDETECT [39], where a higher score (/38) reflects a greater likelihood of neuropathic pain [39]. Body representation was assessed by the Body perception disturbance Scale [32], where a higher score (/57) indicates greater disturbance. Motor and limb ownership related "neglect-like symptoms" were measured by the Neurobehavioral questionnaire [34, 35], where a higher score (/6) indicates a greater severity of "neglect-like" symptoms. The Disabilities of the Arm, Shoulder and Hand questionnaire was used to measure self-reported upper limb disability, where a higher score (/100) indicates a more severe disability [33]. Fear of movement was assessed by the Tampa scale of kinesiophobia, where a higher score (/17) reflects greater fear of movement [36].

Next, all participants completed the MSA pointing task. Participants were seated and rested their head on a chinrest. With their eyes closed, they performed 10 MSA pointing movements using their non-affected/dominant hand, followed by 10 pointing movements with their affected/non-dominant hand. We used a fixed order so that participants with CRPS could become familiar with the task before completing it with their affected hand.

To start a trial, participants placed their index finger on a raised tactile point (~1cm diameter) that was aligned with their body midline and immediately in front of their trunk (the "start location"). A 200 ms auditory cue (800 Hz) signaled that they should fully extend their arm and point their index finger to what felt like straight ahead of their nose, at a comfortable speed. The next trial commenced once a sensor was detected near (i.e. ±2 cm laterally, ±3 cm distally) the start location.

We recorded kinematic data from a sensor placed on the index finger, using an electromagnetic motion capture system (trakSTAR™, 3D Guidance®, Northern Digital Incorporated). For each trial, we calculated angular errors (˚) at movement offset (i.e. once resultant velocity dropped below 50mm/s) from a straight line in the mid-sagittal plane for each trial. Endpoint errors were adjusted to compensate for a calibration error causing a 1.26˚ leftward bias. Errors made towards the affected/non-dominant side of space were coded as negative. See preregistration (https://osf.io/6jpfg/) for full details about the pre-processing of kinematic data, and the clinical and questionnaire measures (available from https://osf.io/t9j52/) used.

### Statistical analyses and inference criteria

We conducted frequentist and Bayesian repeated measures analyses of variance (ANOVAs) to address our hypotheses related to MSA errors and MSA variability. We calculated the mean MSA error and its variability (i.e. SD) from the 10 trials for each participant and for each hand. To address the hypothesis related to the presence of a directional bias in spatial representations, we compared Groups (CRPS, Controls), and Side of Body (affected/non-dominant, non-affected/dominant) on MSA errors. To address the question related to proprioceptive precision, we compared Groups and Side of Body on MSA variability. To see if spatial biases were related to CRPS symptoms and/or disability, we performed correlational analyses of the relationship between MSA, clinical data, and questionnaire measures.

We considered a $p$-value $< 0.05$ as statistically significant for frequentist analyses. For Bayesian analyses we report Bayes Factors (BF) expressed in favour of the alternative hypothesis (i.e. $BF_{10}$), analysed using an uninformed uniform prior. We considered a $BF_{10}$ of 1–3 as inconclusive/anecdotal evidence for the alternative hypothesis, 3–10 moderate evidence, and 10–30 strong evidence [29]. In contrast, we considered a $BF_{10}$ of 1/3-1 as inconclusive/anecdotal evidence for the null hypothesis, 1/10-1/3 moderate evidence, and 1/30-1/10 strong

evidence. We used JASP v0.13.1 [40] for frequentist and Bayesian analyses (see https://osf.io/t9j52/ for data and analysis script).

## Results

### Directional errors

MSA was similar between people with CRPS and controls: we found moderate evidence [29] of no main effect of Group on endpoint errors, $F(1, 33) = 0.01$, $p = 0.971$, $\eta_p^2 < .01$, $BF_{10} = 0.314$. We found a main effect of Side of Body, $F(1, 33) = 14.11$, $p < 0.001$, $\eta_p^2 = .30$, $BF_{10} = 3217.454$ (Fig 1), whereby people made errors towards their affected/non-dominant side when pointing with their affected/non-dominant arm ($M = -4.16˚$, $SE = 1.26$), and errors towards the non-affected/dominant side when using their non-affected/dominant arm ($M = 3.21˚$, $SE = 0.96$). The interaction between Group and Side of Body was not significant, $F(1, 33) = 0.06$, $p = 0.803$, $\eta_p^2 < .01$, with moderate evidence of no effect, $BF_{10} = 0.297$. The results were broadly similar when we re-expressed endpoint errors in terms of left (negative) and right

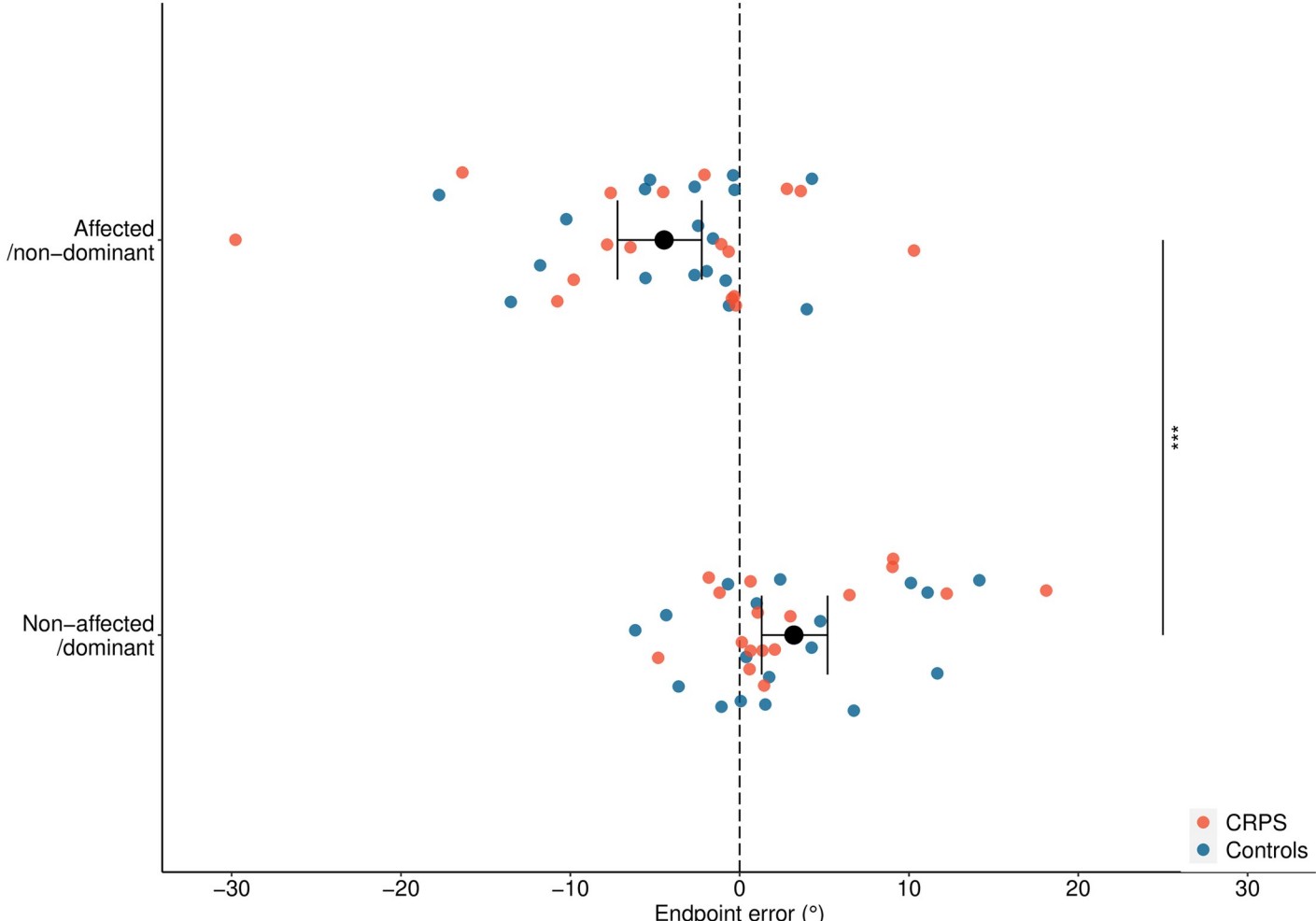

**Fig 1. Manual straight ahead endpoint errors.** Mean endpoint errors in degrees are presented split by Side of Body (affected/non-dominant, non-affected/dominant), for participants with CRPS (orange dots, $n = 17$, $M_{affected} = -4.78˚$, $SD = 8.99$; $M_{non\text{-}affected} = 3.42˚$, $SD = 5.77$), and for control participants (blue dots, $n = 18$, $M_{non\text{-}dominant} = -4.16˚$, $SD = 5.85$, $M_{dominant} = 3.01˚$, $SD = 5.79$). Black dots show mean values for each Side of Body, with bootstrapped 95% confidence intervals (error bars). A negative score indicates errors made towards the affected/non-dominant side. *** $p < 0.001$.

(positive), when clinical measures were included as covariates, and when we excluded participants with comorbid pain conditions. Furthermore, we visually inspected the data split by different CRPS classifications (i.e. NOS, clinical criteria, research criteria), we did not see any indication that these results were shaped by the inclusion of people in these different diagnostic categories.

## Variability

When comparing the intra-individual standard deviations of endpoint errors, we found no significant main effect of Group, $F(1, 33) = 2.96$, $p = 0.095$, $\eta_p^2 = .08$, although the Bayesian analysis was inconclusive, $BF_{10} = 1.10$. There was moderate evidence for no effect of Side of Body, $F(1, 33) = 0.12$, $p = 0.733$, $\eta_p^2 < .01$, $BF_{10} = 0.255$. There was no significant interaction between Group and Side of Body, $F(1, 33) = 2.43$, $p = 0.129$, $\eta_p^2 = .07$, although the Bayesian analysis was inconclusive, $BF_{10} = 0.858$.

## Correlations

We explored correlations between MSA outcomes, clinical data [e.g. CRPS severity; 37], and questionnaire measures [32, 34–36, 39] for participants with CRPS (Fig 2). We did not observe any significant correlations between MSA outcomes and clinical measures. Greater MSA errors towards the non-affected side for the non-affected hand correlated with lower painDETECT scores. The questionnaire measures were highly correlated with each other ($r = .50$ to .86), except for neuropathic-type pain and fear of movement. MSA errors for the affected hand negatively correlated with MSA errors for the non-affected hand, reflecting that individual participants showed similar biomechanical reach behaviour for each hand (i.e. either undershooting or overshooting with each hand). A similar correlation was found for the control participants ($r = -.41$, $p = .091$), albeit non-significant.

## Discussion

We found moderate evidence for no difference between people with CRPS and matched controls on directional MSA endpoint errors. This finding is consistent with previous MSA research [10, 11, 17, 27, 28]. Our study is the first to find statistical evidence in favour of no MSA bias in CPRS relative to controls [41, 42]. We did not find evidence of any differences in MSA variability when comparing between groups or the arm used, consistent with previous research [28]. However, our Bayesian analysis did not provide evidence in favour of no difference.

The only other study to test the equivalence of MSA errors of people with CRPS and pain-free controls was Verfaille and her colleagues [27]. They found inconclusive/anecdotal evidence in favour of no difference when coded relative to the CRPS-affected side of space. Their sample was the same size as ours (i.e. 17 people with CRPS). Although largely comparable, the subtle differences in findings could be due to methodological differences. For instance, we recorded endpoint errors using a motion capture system sensitive to ±1.4 mm. In the study by Verfaille and her colleagues [27], the experimenter visually inspected endpoint errors against a protractor that was precise to 1˚. This difference could have been sufficient to enable our study to detect evidence for the null hypothesis, whereas the findings of Verfaille and her colleagues were inconclusive/anecdotal. It is, however, noteworthy that our findings are largely compatible considering that our study involved people with more chronic CRPS (M duration = 72.65, SD = 41.62) compared to the sample tested by Verfaille and her colleagues (M duration = 13.09, SD = 9.36).

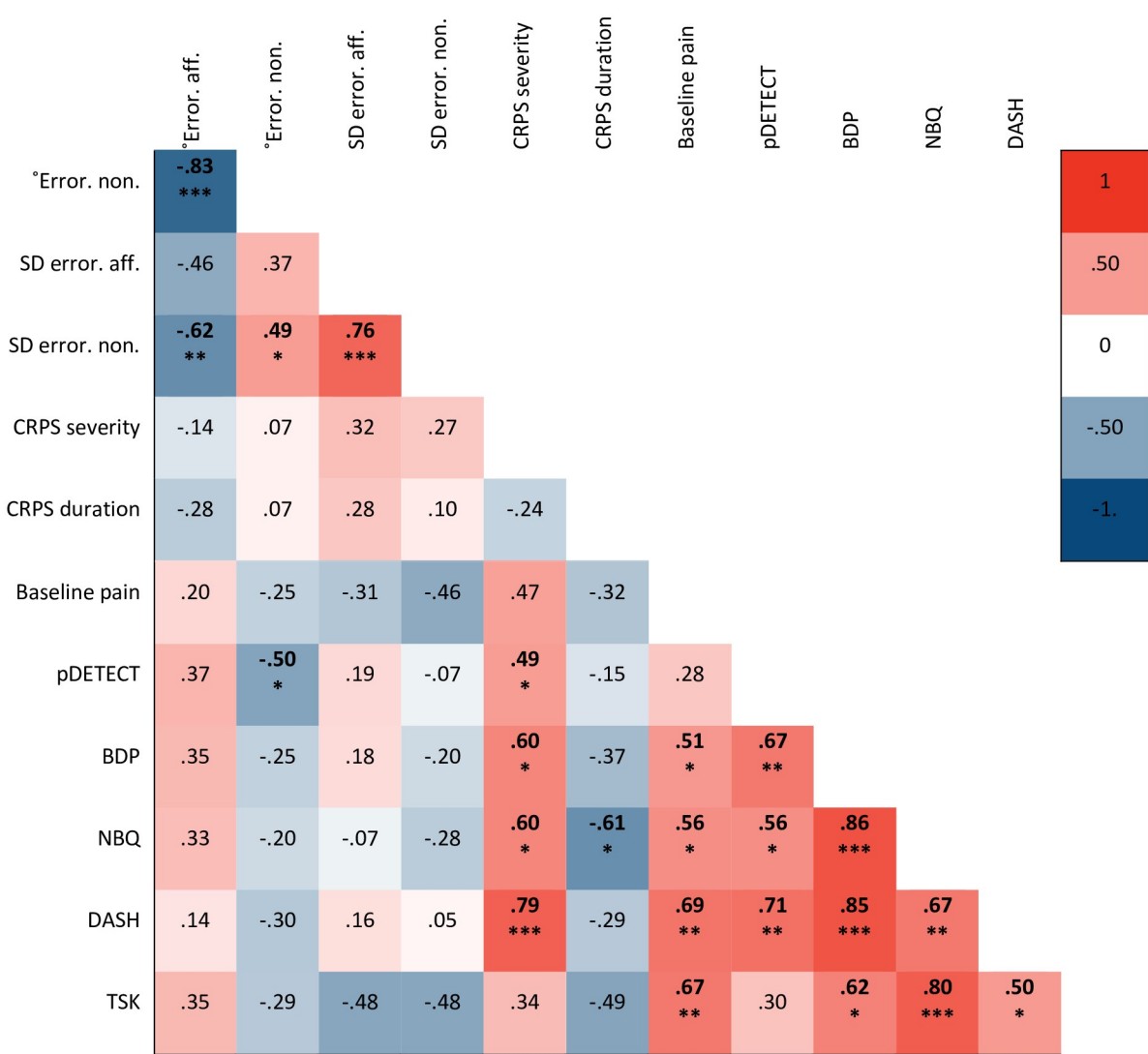

˚Error aff. = endpoint error (˚) for the affected arm. ˚Error non. = endpoint error (˚) for the non-affected arm. SD error, aff. = Intra-individual variability (standard deviations) of endpoint error (˚) for the affected arm. SD error, non. = Intra-individual variability (standard deviations) of endpoint error (˚) for the non-affected arm. pDETECT = painDETECT questionnaire. BDP = Body perception disturbance score. NBQ = Neurobehavioral questionnaire ("neglect-like symptoms"). DASH = The Disabilities of the Arm, Shoulder and Hand questionnaire. TSK = Tampa Scale of Kinesiophobia. *p < 0.05, **p < 0.01, ***p < 0.001

**Fig 2. Correlation matrix for people with upper limb CRPS.** Pearson correlation matrix for people with upper limb CRPS (*n* = 17). CRPS severity was calculated based on the Budapest criteria [1, 2, 37], where a higher score (/16) indicates more signs and symptoms. Significant correlations (i.e. *p* < 0.05) are presented in boldface.

Our study adds to an emerging pattern that MSA has generally been reported as unbiased in CRPS, whereas the evidence is less consistent for the presence or absence of a VSA bias. Discrepancies in the spatial biases in CRPS could relate to the different reference frames (egocentric, allocentric), regions of space (peripersonal, extrapersonal), and/or qualities (goal-

directed, defensive) of the space being tested [3]. For instance, VSA is thought to predominantly rely on egocentric reference frames [14, 18], whereas MSA also involves proprioceptive information [24]. The targets in VSA are typically presented at a distance from participants (e.g. 260 cm away in Verfaille et al. [27]), and thus are in extrapersonal/far space. In MSA, the "target" is a felt position and thus would be considered to be located in personal space and/or imaginary space. These task differences might have given rise to the finding that MSA is unbiased for people with CRPS [17, 27, 28], while some studies suggested that VSA is biased [10–15]. However, it should be noted that the findings of VSA bias in CRPS are not consistent, and that the study that had the largest number of participants did not find any evidence of a bias [n = 53; 16].

Our finding of no difference between the variability of MSA errors of people with CRPS and controls contrasts with reports of bilateral proprioceptive deficits in CRPS on arm position matching tasks [25, 26]. These discrepancies could be due to the reference frames required. MSA is presumed to rely on an egocentric reference frame [24], whereas matching arm position to an external target depends on a combination of egocentric and allocentric reference frames [43]. Recent studies have showed that people with CRPS differ from pain-free controls in how they integrate bodily and visual information [44], and in how they update bodily and spatial representations when interacting with external objects [i.e. tools; 45]. It is possible that this altered integration and/or updating is related to the deficits that are observed on arm matching tasks for people with CRPS. Alternatively, given that our Bayesian analysis was inconclusive about whether the null could be supported, it could be that more subtle differences in MSA variability could be detected in studies involving more participants.

Individual MSA errors did not correlate with clinical variables, or with most questionnaire measures, suggesting that egocentric spatial representations were not related to CRPS symptoms, similar to a recent study of VSA in CRPS [16]. If "neglect-like symptoms" [34, 35] were related to spatial perception biases in CRPS, as in hemispatial neglect, they would correlate with MSA error—we found no such evidence. By contrast, our findings showed that "neglect-like symptoms" were positively correlated with CRPS severity, pain, body representation distortion, upper limb disability, and fear of movement, which is consistent with previous findings [28], and might indicate that neuropsychological changes in CRPS are related to motor-neglect (i.e. underutilisation of a limb/body side that cannot be fully attributed to sensory and/or motor deficits) rather than biased spatial representations [6, 34, 46]. If this were the case, then these findings could inform neuropsychological treatments for CRPS by suggesting that movement/motor-neglect be more effective targets than spatial representations. Therefore, "neglect-like symptoms" are of clinical relevance to CRPS [16], although they appear unrelated to spatial perception biases.

Our study is not without limitations. The sample size meant that our study was only powered to reliably detect large effects, despite having more participants than most previous studies [10, 11, 17] of MSA in CRPS. Although not an aim of the study, this limitation meant that we were not able to explore the direct contribution of factors such as CRPS classification (i.e. not otherwise specified, clinical criteria, research criteria), the presence/absence of painful comorbid conditions, or handedness. The limited power also means that our exploratory correlational analyses should be interpreted with a degree of caution.

## Conclusions

In contrast to potential biases in VSA pointing [10–15], our study corroborates past research showing no directional bias in MSA pointing in CRPS [17, 27, 28]. This finding complements recent findings that show no visuospatial attention bias in people with CRPS [8, 9, 47] by

showing that egocentric spatial representations are also unaffected. The inconsistent findings of spatial biases in CRPS could relate to the different reference frames, regions of space, and/or qualities of the space being tested [3]. We did not find any evidence of impaired proprioception for people with CRPS, which, although consistent with previous MSA findings [28], contrasts findings from position matching task [25, 26]. Therefore, reconciling these discrepant findings is needed to further our understanding of neurocognitive changes in CRPS.

## Author Contributions

**Conceptualization:** Axel D. Vittersø, Janet H. Bultitude.

**Data curation:** Axel D. Vittersø.

**Formal analysis:** Axel D. Vittersø.

**Funding acquisition:** Axel D. Vittersø, Gavin Buckingham, Michael J. Proulx, Janet H. Bultitude.

**Investigation:** Axel D. Vittersø, Antonia F. Ten Brink.

**Methodology:** Axel D. Vittersø, Gavin Buckingham, Janet H. Bultitude.

**Project administration:** Axel D. Vittersø.

**Resources:** Axel D. Vittersø, Antonia F. Ten Brink, Monika Halicka, Janet H. Bultitude.

**Software:** Axel D. Vittersø, Gavin Buckingham.

**Supervision:** Gavin Buckingham, Michael J. Proulx, Janet H. Bultitude.

**Visualization:** Axel D. Vittersø.

**Writing – original draft:** Axel D. Vittersø.

**Writing – review & editing:** Axel D. Vittersø, Gavin Buckingham, Antonia F. Ten Brink, Monika Halicka, Michael J. Proulx, Janet H. Bultitude.

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
