## [Editor Report · Decision Letter 0]

17 May 2021

PONE-D-21-14359

Normal manual straight ahead pointing in Complex Regional Pain Syndrome

PLOS ONE

Dear Dr. Vittersø,

Thank you for submitting your manuscript to PLOS ONE. After careful consideration, we feel that it has merit but does not fully meet PLOS ONE’s publication criteria as it currently stands. Therefore, we invite you to submit a revised version of the manuscript that addresses the points raised during the review process.

Before we can send this manuscript out for review, it needs to meet PLOS ONE's criteria for studies that are closely related to existing work. This study seems to address the same core hypothesis as the authors' 2021 Cortex paper, by again showing no evidence of a visuospatial bias in patients with CRPS. While the current study represents a new method of addressing this question, it does so without adequately discussing the previous work. For example, if these two manuscripts represent two methods addressing the same hypothesis, splitting it into two manuscripts does not remove the need for appropriate multiple-comparison testing.

In addition, given this manuscript's negative/weak results, it needs to discuss whether the sample size was adequate to address the question.

We look forward to receiving your revised manuscript.

Kind regards,

Benjamin A. Philip

Academic Editor

PLOS ONE
---

## [Author Response · Author response to Decision Letter 0]

18 Jun 2021

We would like to thank the editor for his helpful comments on our manuscript. We have revised it following his suggestions, and we believe that the paper is stronger as a result. 

Please see the file labelled "Response to reviewers" for our point-by-point response to the editor's comments.

---

## [Decision Letter · Decision Letter 1]

27 Sep 2021

PONE-D-21-14359R1Normal manual straight ahead pointing in Complex Regional Pain SyndromePLOS ONE

Dear Dr. Vittersø,

Thank you for submitting your manuscript to PLOS ONE. After careful consideration, we feel that it has merit but does not fully meet PLOS ONE’s publication criteria as it currently stands. Therefore, we invite you to submit a revised version of the manuscript that addresses the points raised during the review process.

Please make sure to address - by revisions or explanation - the major concerns of Reviewer #1.

We look forward to receiving your revised manuscript.

Kind regards,

Benjamin A. Philip

Academic Editor

PLOS ONE

Reviewers' comments:

Reviewer's Responses to Questions

**Comments to the Author**

1. If the authors have adequately addressed your comments raised in a previous round of review and you feel that this manuscript is now acceptable for publication, you may indicate that here to bypass the “Comments to the Author” section, enter your conflict of interest statement in the “Confidential to Editor” section, and submit your "Accept" recommendation.

Reviewer #1: (No Response)

Reviewer #2: All comments have been addressed

2. Is the manuscript technically sound, and do the data support the conclusions?

Reviewer #1: Partly

Reviewer #2: Yes

3. Has the statistical analysis been performed appropriately and rigorously? 

Reviewer #1: Yes

Reviewer #2: Yes

4. Have the authors made all data underlying the findings in their manuscript fully available?

Reviewer #1: Yes

Reviewer #2: No

5. Is the manuscript presented in an intelligible fashion and written in standard English?

Reviewer #1: Yes

Reviewer #2: Yes

6. Review Comments to the Author

Reviewer #1: Review Vitterso et al. Normal manual straight ahead pointing in Complex Regional Pain Syndrome.

Dear authors,

sorry that this took a while to do this review.

In the paper you present a case series of 17 patients with CRPS in contrast to control subjects that were age and gender matched.

There were no findings that indicate a difference between the 2 groups in the manual straightahead while there are ambiguous evidences for deviations towards the unaffected side („neglect- like“) from other studies.

You present an established paradigm and adeaquately point out the difference between the different reference frames and therefor the reasoning for choosing the presented paradigm over VSM. I wish we could compare the results of VSM to MSA in the same subjects to compare the methods which would give us more insights on the different mechanisms of the two tasks and the involved pathophysiologic basis. But anyways from most of the studies there is no effect in VSM as there is no effect in MSA. Yet, your paradigm is not very innovative from my point of view but the statistical approach indeed is (to be honest: I think analysing all the negative studies on pointing tasks in a similar way as you did might lead to comparable results).

I do not fully agree with the notion that involving the sensory system in terms of proprioception in a pointing task will improve the egocentricness of this task as sensory symptoms and muscular changes affecting proprioception are part of the diagnosis of CRPS and impaired hand size estimation (Petz et al. 2011) is also a symptom of CRPS as well as loss of proprioceptive accuracy in positioning tasks (Lewis 2010, which you also cite but maybe interpret it in a more „centralistic“ way than I do), so I would argue against any positive finding of the MSA- pointing showing a between groups effect as aequivalent of egocentric representation which is not the case in this study so may be not a problem but maybe you should weigh the effects of sensory symptoms in the reasoning of choosing your paradigm and reconsider the CRPS- unbiased egocentricness of it always keeping in mind that we are seaching for central representation meachnisms and not peripheral effects of CRPS.

This is in some way adressed in lines 139 ff. but in the paragraphs before concepts get a bit mixed up.

Though not really large, the group of patients is larger than in comparison to other studies with manual straightahead pointing tasks but on the other hand the effects one would expect from those studies are not that large that I can imagine that those studies went into power calculation for the study presented here.

The choosing of sample sizes for pragmatic reasons is really honest but does not seem very scientific to me, which is my ponit of biggest concern and which limits the impact of the findings a lot ( see also next paragraph). Anyways you present not only the absence of evidence for a difference between groups (which would be most likely of doubt because of the number of subjects) but the actual evidence for absence of difference between groups. So leaving the generally sparse (and certainly underpowered from a statistical point of view) number of subjects out of the scope, what seems like a statistical trick at first sight, might indeed add some crucial point to the scientific debate (if maybe it is only to stop doing pointing tasks with our patients).

I personally think that scientific studies for CRPS should not include subjects with the diagnosis CRPS NOS but only CRPS –patients at least fulfilling clinical criteria. I would be interested in the results of the study when leaving CRPS NOS- patients out of the statistical consideration. It might even improve the results due to loss of variance or due to loss of power lower the effect size.

I would also not include patients with other pain disorders such as FMS, knee damage or migraine in CRPS- studies.

Did you test for the effect of handedness in your statistical model and the interaction term of affected * dominance? What we know from other studies is that people tend to bisect a room more on the left side so I think that we need to account for the amount of pseudoneglect, which is more pronounced in right handed subjects than in left handed subjects as baseline correction of the statistcal model or try to explore the interactionterms.

What are your thoughts about the negative significant correlation between the error of the affected side and the error of the affected side? Maybe they can elaborate on that in your discussion a bit more.

Concerning line 240 ff. : I think that we learned a lot about the influence of anxiety and kinesiophobia in CRPS in the recent years and their interrelation with peripersonal space and extrapersonal allodynia. Maybe we need to consider not only allocentric/egocentric reference frame but also the nearness of the task target to the peripersonal space. You adressed that in your conclusions but it might be a crucial point here.

Some minor non- scientific remarks:

Line 40-41 this sentence is a bit unclear, what is meant by „and according to the arm used“, I think this refers to „We compared endpoint errors“, but the sentence is a bit odd.

Lines 43 and 44 Groups is spelled with a capital G.

Line 48 „This study is consistent …“

Line 58 „although see“  rest of the sentence is missing

Line 66 „VSA is thought to reflect any lateral shifts of…“  missing word

Line 106 CRSP  CRPS

Line 133 Groups and Side of Body spelled with capitals

Line 135 „we would expect to see a greater“ (the „a“ is a word to much)

Table 1 migraines  migraine

Line 185 questionnaires measures („s“ or „measures“ to much)

Line 229 Groups and Side of Body spelled with capitals

Line 246 and 247 Groups and Side of Body spelled with capitals

Line 251 Groups and Side of Body spelled with capitals

So in general I think this is a interesting footnote to the scientific debate rather than a very innovative fully grown paper. My greatest concern is the selection of subjects for your study and the number of subjects in your study which can not be improved post hoc but maybe should be adressed in a „limitations“- section.

Best regards

Reviewer #2: The authors have now addressed all concerns sufficiently and I now suggest to accept the manuscript as it stands now.

7. PLOS authors have the option to publish the peer review history of their article (what does this mean?). If published, this will include your full peer review and any attached files.

Reviewer #1: No

Reviewer #2: **Yes: **Martin Lotze

---

## [Author Response · Author response to Decision Letter 1]

2 Nov 2021

Please see the attached file labeled "Response to reviewers_Rev1".

---

## [Decision Letter · Decision Letter 2]

7 Dec 2021

Normal manual straight ahead pointing in Complex Regional Pain Syndrome

PONE-D-21-14359R2

Dear Dr. Vittersø,

We’re pleased to inform you that your manuscript has been judged scientifically suitable for publication and will be formally accepted for publication once it meets all outstanding technical requirements.

Kind regards,

Dario Ummarino, Ph.D.

Senior Editor

PLOS ONE

Reviewer's Responses to Questions

**Comments to the Author**

1. If the authors have adequately addressed your comments raised in a previous round of review and you feel that this manuscript is now acceptable for publication, you may indicate that here to bypass the “Comments to the Author” section, enter your conflict of interest statement in the “Confidential to Editor” section, and submit your "Accept" recommendation.

Reviewer #1: All comments have been addressed

Reviewer #2: All comments have been addressed

2. Is the manuscript technically sound, and do the data support the conclusions?

Reviewer #1: Yes

Reviewer #2: Yes

3. Has the statistical analysis been performed appropriately and rigorously? 

Reviewer #1: Yes

Reviewer #2: Yes

4. Have the authors made all data underlying the findings in their manuscript fully available?

Reviewer #1: Yes

Reviewer #2: Yes

5. Is the manuscript presented in an intelligible fashion and written in standard English?

Reviewer #1: Yes

Reviewer #2: Yes

6. Review Comments to the Author

Reviewer #1: Dear authors, thank you for the in depth revision of your manuscript and the comprehensive answers to my questions/ remarks as well as pointing out some aspects, that I might have overseen in the debate about neglect-like symptoms in CRPS and hints to some additional interesting articles to read, which again took me a while.

You are completely right with time and financial constraints in science and I highly appreciate your honesty in this point. You are completely right also choosing a realistic group of CRPS- patients (i.e. with comorbidities and different criteria) keeping that point in mind and as you show in your review (rev-fig 1, rev-fig2) choosing the sample in the way you did does not really affect variance (maybe variance is a bit higher in the patients with CRPS research criteria while pointing with the affected hand). So I am fine with your revision on that point and I agree that statistical tests between the subsets do not really make sense given the sparse cell numbers. I also agree with your elaborations on sample size and power in the revised manuscript and the conclusions drawn from the datasplit.

Thank you also for sharing your thoughts about the negative significant correlations and adding those to the manuscript. I think without overinterpreting it, that this might be an interesting finding.

Overall I think that you adressed all my concerns and cherry-pickings in an apropriate and comprehensive way not only scratching the surface. Thanks a lot for that effort.

Reviewer #2: (No Response)

7. PLOS authors have the option to publish the peer review history of their article (what does this mean?). If published, this will include your full peer review and any attached files.

Reviewer #1: No

Reviewer #2: **Yes: **Martin Lotze

---

## [Editor Report · Acceptance letter]

10 Dec 2021

PONE-D-21-14359R2 

Normal manual straight ahead pointing in Complex Regional Pain Syndrome 

Dear Dr. Vittersø:

I'm pleased to inform you that your manuscript has been deemed suitable for publication in PLOS ONE. Congratulations! Your manuscript is now with our production department. 

Kind regards, 

on behalf of

Dr. PLOS Manuscript Reassignment 

Staff Editor

PLOS ONE